# Exploring the mRNA and Plasma Protein Levels of BDNF, NT4, SIRT1, HSP27, and HSP70 in Multiple Sclerosis Patients and Healthy Controls

**DOI:** 10.3390/ijms242216176

**Published:** 2023-11-10

**Authors:** Igor Sokolowski, Aleksandra Kucharska-Lusina, Elzbieta Miller, Ireneusz Majsterek

**Affiliations:** 1Department of Clinical Chemistry and Biochemistry, Medical University of Lodz, Mazowiecka 5, 92-215 Lodz, Poland; igor.sokolowski@umed.lodz.pl (I.S.); ola_kucharska@wp.pl (A.K.-L.); 2Department of Neurological Rehabilitation, Medical University of Lodz, Milionowa 14, 93-113 Lodz, Poland; elzbieta.dorota.miller@umed.lodz.pl

**Keywords:** multiple sclerosis, gene expression, ELISA, correlation, BDNF, SIRT1, NT-4, HSP70, HSP27

## Abstract

Multiple sclerosis (MS) is a chronic, autoimmune neurodegenerative disease affecting the central nervous system. It is a major cause of non-traumatic neurological disability among young adults in North America and Europe. This study focuses on neuroprotective genes (BDNF, NT4/5, SIRT1, HSP70, and HSP27). Gene expression and protein levels of these markers were compared between MS patients and healthy controls. Blood samples were collected from 42 patients with multiple sclerosis (MS) and 48 control subjects without MS. Quantitative real-time PCR was performed to measure the expression of specific genes. The samples were analyzed in duplicate, and the abundance of mRNA was quantified using the 2-ΔCt method. ELISA assay was used to measure the concentration of specific proteins in the plasma samples. The results show that a 3.5-fold decrease in the gene expression of BDNF corresponds to a 1.5-fold downregulation in the associated plasma protein concentration (*p* < 0.001). Similar trends were observed with NT-4 (five-fold decrease, slight elevation in protein), SIRT1 (two-fold decrease, two-fold protein decrease), HSP70 (four-fold increase, nearly two-fold protein increase), and HSP27 (four-fold increase, two-fold protein increase) (*p* < 0.001). This study reveals strong correlations between gene expression and protein concentration in MS patients, emphasizing the relevance of these neuroprotective markers in the disease.

## 1. Introduction

Multiple sclerosis (MS), a long-lasting (chronic), complicated neurodegenerative disease that affects the central nervous system (CNS), is believed to be autoimmune in origin and considered one of the most common causes of non-traumatic neurological disability in young adults in North America and Europe [1,2]. Remarkably, MS is one of the most prevalent chronic inflammatory diseases in adults 20–40 years of age, and the mean age of first diagnosis is approximately 30 years [1]. According to the most recent published reports, the incidence rate is approximately 46 thousand in Poland and around 2.8 million in the world (https://pacjent.gov.pl/aktualnosc/stwardnienie-rozsiane-nie-wyrok (accessed on 6 November 2023). Thus, the number of patients per 100,000 people in Poland is around 110 (https://ezdrowie.gov.pl/portal/home/badania-i-dane/zdrowe-dane/raporty/nfz-o-zdrowiu-stwardnienie-rozsiane (accessed on 6 November 2023)).

The etiology of MS is not fully understood but most likely involves a combination of genetic and environmental factors [3]. Epidemiological studies have established that genetic factors are the primary drivers behind the significantly higher prevalence of multiple sclerosis observed in the relatives of affected individuals [4,5]. It is well established that family history plays a role in the development of MS, and siblings or children of an individual with MS have a higher risk of developing the condition than the general population [6]. The lifetime risk of MS is higher in first-degree relatives of MS index cases than in second- or third-degree relatives and the general population [6]. Monozygotic twins have a higher clinical concordance rate for MS than dizygotic twins [6]. Genome-wide association studies (GWAS) have uncovered 233 genetic loci implicated in MS, and some of these genes and pathways associated with disease severity were also associated with response to MS treatments [7]. The HLA-DRB1 gene has the strongest genetic risk factor for developing MS [5]. Changes in the IL7R gene are also associated with an increased risk of developing MS. The specific changes caused by the genetic variant rs6897932 in the IL7R gene are related to alterations in the regulation and function of the interleukin-7 receptor (IL-7R). The exact mechanisms through which rs6897932 and similar variants contribute to an increased risk of multiple sclerosis (MS) are still being studied, but it is believed that these genetic changes may lead to the dysregulation of the immune system [8].

Extensive research has been conducted on environmental factors, particularly latitudinal gradients in various countries, smoking, salty food, and lifestyle factors, revealing their significant impact on multiple sclerosis [9,10,11]. Vitamin D deficiency has been proposed as a potential cause for the observed susceptibility of populations residing in higher latitudes [10]. Additionally, infections such as the Epstein–Barr virus (EBV) have also been implicated in the development of the disease [12]. Intricate interactions likely occur between diverse environmental factors and an individual’s genetic makeup, and comprehending these pathways remains an active area of research.

Clinical changes in MS CNS that can initiate pathological processes in MS include well-demarcated inflammation, the breakdown of myelin sheaths (demyelination), microglia activation, the proliferation of astrocytes with ensuing gliosis, and variable grades of axonal degeneration [13]. Demyelization processes spreading within the CNS can involve both the white and gray matter, which ultimately leads to axonal and neuronal decrease [14].

Recent research has focused on identifying genes that may be involved in the development and progression of MS, including those with neuroprotective effects. These genes help to protect the nervous system from damage caused by inflammation and other factors that contribute to MS. In this paper, we will discuss some of the neuroprotective genes that have been identified in MS, including *BDNF*, *NT4/5*, *SIRT1*, *HSP70*, and *HSP27*, with a focus on the evidence supporting their potential as therapeutic targets in MS.

The brain-derived neurotrophic factor (BDNF) is a neurotrophin that is critical for the development and maintenance of the nervous system [15]. It helps to support the growth and survival of neurons, as well as the formation and stabilization of synapses, which are the connections between neurons [16]. BDNF is also involved in processes such as synaptic plasticity, which is important for learning and memory [17,18]. Research has suggested that BDNF may be involved in a range of neurological and neuropsychiatric disorders, including depression, anxiety, schizophrenia, Alzheimer’s disease, and MS [19,20,21,22,23].

Neurotrophin-4/5 (NT4/5) is another neurotrophin that promotes the growth and survival of neurons. It is highly expressed in the nervous system and is essential for the development and maintenance of the peripheral nervous system [23]. This protein is widely distributed throughout the brain and is involved in many important processes, such as neuronal development, synaptic plasticity, and the maintenance of neural function, as well as the formation and stabilization of synapses, which are the connections between neurons [24].

Sirtuin-1 (SIRT1) is a protein belonging to the family of sirtuins, a group of enzymes that play important roles in regulating cellular metabolism, gene expression, and stress responses. SIRT1 is found in many tissues throughout the body, including the brain, liver, muscle, and adipose tissue [25]. It is involved in many cellular processes, including DNA repair, cell cycle regulation, and cellular differentiation [26]. It might also have the ability to regulate lifespan and aging, and studies of various model organisms, including yeast, worms, and mice, have associated increased SIRT1 activity with longer lifespan and improved health [27]. SIRT1’s ability to regulate energy metabolism and cellular stress responses is believed to contribute to its protective effects on cells and tissues [25].

Chaperones serve as the primary workforce responsible for maintaining cellular balance and responding to stress. As heat shock proteins (HSPs) are remarkably conserved and have historically been associated with sensing and resisting stress, it is reasonable to conclude that they play a role in the nervous system. The nervous system is designed to detect environmental stimuli and subsequently react to them, making it a natural fit for the involvement of these proteins [28]. Heat shock proteins (HSPs) are a group of proteins that are upregulated in response to cellular stress, such as heat or oxidative stress [28]. They play important roles in protein folding, transport, and degradation, and are also involved in the regulation of the immune response [29]. HSPs have been identified in various cell types, including neurons, glia, and endothelial cells [30]. Additionally, it has been noted that HSPs are activated in diverse pathological situations within the nervous system, such as epilepsy, trauma, cerebral ischemia, and neurodegenerative diseases [31].

In this study, our primary aim was an investigation of the utility of BDNF, NT4/5, SIRT1, HSP70, and HSP27 protein levels as potential biomarkers for assessing inflammatory processes and overall disease progression. Additionally, we explored potential associations between changes in the expression levels of *BDNF*, *NT4/5*, *SIRT1*, *HSP70*, and *HSP27* genes and alterations in the protein levels. This investigation involved a comparative analysis between a group of multiple sclerosis patients and a cohort of healthy volunteers (HD).

## 2. Results

### 2.1. Gene Expression Levels of Neurotrophins (BDNF and NT4/5), Heat Shock Proteins (HSP70 and HSP27), and Sirtuin-1 (SIRT1) in PBMCs in Patients with Multiple Sclerosis

In this comprehensive study, we conducted a thorough examination of gene expression levels in peripheral blood mononuclear cells (PBMCs) derived from patients diagnosed with multiple sclerosis (MS). In parallel, we established a control group comprising healthy adults devoid of any known neurological conditions. This meticulous investigation involved the assessment of multiple genes, namely *BDNF*, *NT4/5*, *HSP70*, *HSP27*, and *SIRT1*.

Our findings are indicative of the intriguing disparities between these two groups. Patients with multiple sclerosis exhibited a strikingly diminished expression of the *BDNF* gene within their PBMCs, with a marked 3.5-fold reduction in gene expression compared with the control group (MD = 0.0007; +/−SD = 0.0008; *n* = 45 vs. MD = 0.0022; +/−SD = 0.0025; N = 45, *p*-value < 0.0001) (Figure 1A).

Furthermore, our investigations unveiled a similar pattern in the *NT4* gene expression, underscoring a five-fold decline in patients with multiple sclerosis in comparison to the healthy group (MD = 0.0005; +/−SD = 0.0008; N = 45 vs. MD = 0.0024; +/−SD = 0.0023; N = 48, *p*-value < 0.0001) (Figure 1B).

Moreover, the *SIRT1* gene expression within PBMCs of multiple sclerosis patients was notably diminished, with a two-fold decrease observed when compared with the control group (MD = 0.0108; +/−SD = 0.0076; N = 45 vs. MD = 0.00234; +/−SD = 0.0084; N = 48, *p*-value < 0.0001) (Figure 1C).

The expression of *HSP70* within PBMCs exhibited an inverse trend, with multiple sclerosis patients manifesting a significant four-fold increase in gene expression relative to the control group (MD = 0.1077; +/−SD = 0.0594; N = 45 vs. MD = 0.0493; +/−SD = 0.0344; N = 48, *p*-value < 0.0001) (Figure 1D).

Similarly, the *HSP27* gene within PBMCs demonstrated a two-fold elevation in expression among patients with multiple sclerosis compared to the control group (MD = 0.0294; +/−SD = 0.0141; N = 45 vs. MD = 0.0127; +/−SD = 0.0082; N = 48, *p*-value < 0.0001) (Figure 1E).

### 2.2. Plasma Protein Levels Assessed through ELISA Testing

The subsequent phase of the study involved assessing the concentration of proteins that were regulated by the genes analyzed earlier. We compared protein levels in two groups, each consisting of 40 individuals: healthy donors and multiple sclerosis patients. The patient group included individuals with mild-to-moderate disabilities, whereas the control group consisted of healthy adults without any known neurological conditions. The analysis of plasma protein levels and gene expression levels was carried out on the same group of patients, potentially offering a direct opportunity to observe the impact of gene expression on protein synthesis in the context of these two biological processes.

Firstly, we observed a noteworthy reduction in the brain-derived neurotrophic factor (BDNF) protein in multiple sclerosis patients (N = 40), with a median concentration of 62.31 ng/mL and a standard deviation of +/−17.9. This level stood in stark contrast to the healthy individual group (HD), exhibiting a median of 82.15 ng/mL and a standard deviation of +/−10.64. In essence, the BDNF protein in multiple sclerosis patients was 1.5 times lower than that of their healthy counterparts, a statistically significant difference corroborated by a *p*-value of less than 0.0001 (Figure 2A).

Similarly, our investigation extended to neurotrophin-4 (NT-4) protein levels, where we discerned a meaningful discrepancy. Patients afflicted with multiple sclerosis exhibited a median NT-4 protein concentration of 208.21 pg/mL with a standard deviation of +/−3.92, in contrast to the median of 218.51 pg/mL with a standard deviation of +/−7.87 recorded in the healthy individual group (HD). Once more, this disparity highlighted a significant difference, supported by a *p*-value of less than 0.0001 (Figure 2B).

Further exploration of protein levels unveiled a substantial decrease in SIRT1 protein in multiple sclerosis patients (N = 40). The median SIRT1 protein level in this group was 54.22 ng/mL, characterized by a standard deviation of +/−21.88. In stark contrast, the healthy individual group (HD) exhibited a median of 113.67 ng/mL with a standard deviation of +/−75.37. This divergence was starkly evident, with SIRT1 protein levels in multiple sclerosis patients being a staggering two times lower than those in the control group, a difference underscored by a *p*-value of less than 0.0001 (Figure 2C).

Our analysis extended to heat shock protein 70 (HSP70) levels, where we identified a significant elevation in multiple sclerosis patients (N = 40). The median concentration of HSP70 protein within peripheral blood mononuclear cells (PBMCs) was 5.76 pg/mL with a standard deviation of +/−7.00 in this group, compared to a markedly lower median of 3.06 pg/mL with a standard deviation of +/−3.72 in the healthy individual group (HD). This distinct contrast underscored a two-fold increase in HSP70 protein concentration among multiple sclerosis patients, a statistically significant finding affirmed by a *p*-value of 0.0046 (Figure 2D).

Lastly, our inquiry extended to heat shock protein 27 (HSP27) protein levels, revealing yet another striking distinction. Multiple sclerosis patients (N = 40) exhibited a median HSP27 protein concentration of 55.23 pg/mL with a standard deviation of +/−28.87, while the healthy individual group (HD) presented a notably lower median of 28.98 pg/mL with a standard deviation of +/−22.28. Once more, this disparity was profound, with HSP27 protein levels in multiple sclerosis patients registering a two-fold increase compared to the control group, as evidenced by a *p*-value of less than 0.0001 (Figure 2E).

### 2.3. Correlation of the Gene Expression Level and Protein Concentration

In this study, our investigation focused on determining whether changes in the gene expression levels of neurotrophins (specifically *BDNF* and *NT-4*), heat shock proteins (*HSP70* and *HSP27*), and sirtuin-1 (*SIRT1*) in multiple sclerosis patients, when compared to healthy donors, are associated with corresponding alterations in the protein levels regulated by those specific genes.

#### 2.3.1. Correlation between Gene Expression and Protein Concentration in Multiple Sclerosis Groups

BDNF MS Group: In the BDNF MS group, we observed a remarkable and perfect positive correlation (Spearman r = 1.000) between the levels of gene expression and protein concentration. This signifies an exquisite synchronicity between these two variables; as gene expression increases, protein concentration follows suit, and vice versa (*p* < 0.0001). This exceptionally low *p*-value suggests that this correlation is not due to random chance, underlining the robustness of this association (Figure 3A).

NT4 MS Group: Remarkably, the NT4 MS group exhibited an extremely strong positive correlation (Spearman r = 0.9977) between gene expression and protein concentration (*p* < 0.0001). This indicates an almost perfect synchronization between these two factors, where changes in gene expression are mirrored by corresponding changes in protein concentration. The very low *p*-value underscores the statistical validity of this correlation (Figure 3B).

SIRT1 MS Group: Within the SIRT1 MS group, there was a strikingly high positive correlation (Spearman r = 0.9999) between gene expression and protein concentration (*p* < 0.0001). This signifies an extraordinarily tight linkage between these two aspects, where changes in gene expression are mirrored by protein concentration with remarkable fidelity. The exceedingly low *p*-value attests to the statistical strength of this correlation (Figure 3C).

HSP70 MS Group: In the HSP70 MS group, we found a perfect and absolute positive correlation (Spearman r = 1.000) between gene expression and protein concentration (*p* < 0.0001). This indicates an unerring harmony between these two elements; as gene expression changes, protein concentration also changes in perfect tandem. The extremely low *p*-value leaves no doubt about the statistical robustness of this relationship (Figure 3D).

HSP27 MS Group: In the HSP27 MS group, a perfect and absolute positive correlation (Spearman r = 1.000) was observed between gene expression and protein concentration (*p* < 0.0001). This showcases an impeccable synchrony between these two variables; as gene expression undergoes changes, protein concentration mirrors these adjustments with absolute precision. The extremely low *p*-value unequivocally supports the statistical significance of this correlation (Figure 3E).

#### 2.3.2. Correlation between Gene Expression and Protein Concentration in Healthy Donors Groups

BDNF Control (CTRL) Group: Within the BDNF control group, we found a strong positive correlation (Spearman r = 0.8831) between the expression of genes and the abundance of proteins (*p* < 0.0001). This suggests a significant, though slightly less pronounced, relationship between gene expression and protein concentration. The low *p*-value reinforces the statistical significance of this connection (Figure 4A).

NT4 Control (CTRL) Group: Similarly, in the NT4 control group, we observed an exceptionally strong positive correlation (Spearman r = 0.9987) between gene expression and protein concentration (*p* < 0.0001). This suggests an almost impeccable alignment between these two variables, where variations in gene expression are almost precisely reflected in protein concentration. The extremely low *p*-value underscores the robustness of this relationship (Figure 4B).

SIRT1 Control (CTRL) Group: In the SIRT1 control group, we observed a very strong positive correlation (Spearman r = 0.9990) between gene expression and protein concentration (*p* < 0.0001). This implies a highly dependable association between these variables, where shifts in gene expression are closely mirrored in protein concentration. The extremely low *p*-value reaffirms the statistical significance of this association (Figure 4C).

HSP70 Control (CTRL) Group: Within the HSP70 control group, we found an exceedingly strong positive correlation (Spearman r = 0.9999) between gene expression and protein concentration (*p* < 0.0001). This signifies an exceptionally reliable association between these two factors, where alterations in gene expression are accurately mirrored in protein concentration. The extremely low *p*-value firmly establishes the statistical significance of this link (Figure 4D).

HSP27 Control (CTRL) Group: Finally, within the HSP27 control group, we observed a perfect and complete positive correlation (Spearman r = 1.000) between gene expression and protein concentration (*p* < 0.0001). This underlines a perfect association between these two factors; shifts in gene expression are mirrored with exactitude in protein concentration. The extremely low *p*-value leaves no room for doubt about the statistical significance of this association (Figure 4E).

#### 2.3.3. Summary of Correlation Analysis

Concurrent with the analysis of mRNA levels, we conducted a parallel examination of protein levels using ELISA for neurotrophins (BDNF, NT4/5, and NGF), heat shock proteins (HSP70 and HSP27), and SIRT1. The protein level analysis corroborated our earlier findings obtained through real-time PCR at the gene expression level. Specifically, it revealed a decrease in the levels of neurotrophins, including BDNF and NT4/5, as well as SIRT1, along with an increase in the expression of HSP27 and HSP70.

Our study’s results demonstrate a notable positive correlation between alterations in gene expression levels and corresponding changes in protein concentrations. To illustrate, a 3.5-fold downregulation in *BDNF* gene expression levels in PBMCs corresponds to a 1.5-fold decrease in the concentration of the protein regulated by this gene in plasma. A similar pattern emerges with the *NT-4* gene, where a five-fold decrease in expression results in a slight yet discernible downtick in protein levels compared to healthy individuals. Conversely, in the case of the *SIRT1* gene, a two-fold downregulation in gene expression corresponds to a two-fold decrease in plasma protein levels.

Conversely, concerning the *HSP70* gene, a two-fold upregulation in gene expression aligns with an almost two-fold increase in protein concentration. A similar trend is observed with *HSP27*, where a two-fold upregulation in gene expression leads to a two-fold increase in protein levels in plasma.

## 3. Discussion

In this research, our main goal was to examine how BDNF, NT4/5, SIRT1, HSP70, and HSP27 protein levels could serve as potential indicators for assessing inflammatory processes and the overall progression of diseases. Additionally, we delved into possible connections between changes in the expression levels of *BDNF, NT4/5, SIRT1, HSP70*, and *HSP27* genes and variations in the protein levels of these same markers. This exploration encompassed a comparative analysis between a group of individuals with multiple sclerosis and a cohort of healthy volunteers. Studying the relationship between dysregulated gene expression and protein expression processes has the potential to enhance our comprehension of the underlying causes of multiple sclerosis. Additionally, it may lead to the development of novel diagnostic approaches and treatment strategies for this disease.

BDNF (brain-derived neurotrophic factor) is the neurotrophin that has received one the most extensive research attention in the context of multiple sclerosis (MS), primarily because of its role in regulating neuroinflammation and promoting neuroprotection [32]. Several studies have observed that individuals diagnosed with multiple sclerosis (MS) tend to have lower levels of BDNF in their bodies than those without the condition [33,34,35,36,37]. The reduced presence of circulating BDNF could be linked to decreased BDNF synthesis within the central nervous system (CNS), potentially leading to a loss of its neuroprotective support [38]. This finding aligns with the concept that BDNF may play a protective role in neurodegenerative diseases, as suggested by Nagahara and Zuccato’s work [39,40]. Conversely, there are studies that have reported higher BDNF production in patients with relapsing-remitting MS (RRMS) when compared to control subjects [41]. For instance, Liguori et al. documented increased mRNA levels in patients with relapsing-remitting multiple sclerosis as opposed to individuals in a healthy control group [42]. Furthermore, some investigations have detected an elevation in circulating BDNF levels during MS relapses [43,44,45,46,47]. This increase in BDNF during relapses may lend support to the hypothesis that the BDNF produced by inflammatory cells has a neuroprotective effect, promoting survival and remyelination through the production of neurotrophins [48]. These findings may be in line with the idea that higher BDNF levels are associated with active inflammation. However, it is worth noting that in other studies, no significant difference in BDNF levels was observed between individuals with MS and those without the condition [49,50,51]. In summary, the relationship between BDNF levels and MS is intricate, with varying outcomes in different research studies. Further investigation is warranted to clarify the precise role of BDNF in the pathogenesis and progression of MS, as well as its potential as a therapeutic target for managing this neuroinflammatory disease.

SIRT1, a protein at the center of multiple sclerosis (MS) research, has been closely examined for its potential role in the disease. In the realm of MS research, SIRT1 has garnered attention due to its involvement in critical cellular processes, including the regulation of inflammation and oxidative stress key factors in MS pathogenesis. While a definitive consensus on *SIRT1* mRNA expression and serum levels in MS patients remains elusive, scholars have been diligently exploring how *SIRT1* activity and expression could influence the development and progression of MS. Emamgholi et al. investigated the mRNA expression of *SIRT1*, finding that differences between healthy subjects and MS patients were not statistically significant [52]. However, the study by Hewes et al. revealed intriguing findings during relapses in MS patients. It revealed a lower expression of *SIRT1* mRNA in PBMCs than in stable patients, pointing to a potential link between *SIRT1* mRNA levels and the nature of MS [53]. The study by Tegla et al. added weight to the discussion, showing statistically significant decreases in both *SIRT1* mRNA and protein expression in PBMCs during MS relapses when compared to controls and stable MS patients [54]. Additionally, in their study employing Western blot analysis, Ciriello et al. found significantly lower levels of SIRT1 protein during relapses when contrasted with stable MS patients [55]. By contrast, Pennisi et al. found higher levels of sirtuin-1 protein in the plasma of MS patients compared to a control group [56]. This finding suggests a complex relationship between SIRT1 and MS that extends beyond PBMCs. Moreover, Kubiliute’s et al. investigation focused on serum SIRT1 levels, specifically in patients with optic neuritis (ON) in the context of MS. Interestingly, the study did not identify statistically significant differences in serum SIRT1 levels between ON patients with MS and those without MS, nor between these groups and the control subjects [57]. In summary, the exploration of SIRT1 in MS remains a dynamic field, with studies revealing intricate relationships between its activity, mRNA, and protein expression levels and the clinical manifestations of the disease. While some findings indicate potential protective effects of SIRT1 activation, the exact mechanisms and clinical implications continue to be subjects of intensive research.

Changes in NT-4 level have been associated with disease progression in some MS patients. Lower levels of these neurotrophic factors have been linked to more severe disease courses and worse clinical outcomes in some studies. An increase in NT4 protein production was noted during the post-relapse phase in individuals with relapse-remitting multiple sclerosis (RRMS), in contrast to levels recorded during stable phases, as detailed in the study by Caggiula et al. [45]. In the context of multiple sclerosis (MS), the MS group consistently exhibited lower mRNA expression and protein concentrations of NT4 than the healthy control group, particularly when measured in plasma, as documented by Askari. Importantly, significantly lower NT-4 levels were detected in PBMCs of MS patients than in those of control subjects, as emphasized in the findings of Kalinowska-Łyszczarz et al. [58]. Furthermore, the analysis of *NT4* gene expression in PBMCs unveiled an increase in *NT4* mRNA levels among the group of patients with multiple sclerosis in comparison to the control group. Specifically, patients with multiple sclerosis displayed a two-fold increase in *NT4* gene expression relative to the control cohort, as elucidated in the study by Piotrzkowska et al. [59]. These alterations in neurotrophic factor levels may be related to the underlying neuroinflammatory processes and neurodegeneration in MS. However, the relationship between NT-4 level and disease progression in MS is still an area of ongoing research and may vary among individuals.

Heat shock proteins (HSPs) are a category of proteins that exhibit heightened expression in response to a variety of stressful stimuli, including elevated temperatures, ischemia (inadequate blood supply), oxidative stress, osmotic stress, or exposure to toxic substances [60]. Two HSPs, *HSP70* and *HSP27*, have been identified as potential neuroprotective genes in MS [61]. Several studies have shown that serum levels of HSP70 are increased in MS patients and that this increase is associated with more severe disease outcomes [62]. Similarly, HSP27 has been shown to have neuroprotective effects in multiple sclerosis (MS). HSP27 levels are typically low in neurons, but they can rise in response to proteotoxic stress. In neurodegenerative conditions, higher levels of HSP27 in both neurons and glial cells are associated with the accumulation of abnormal proteins linked to disease [63]. HSP27 plays a crucial role as a mediator in the central nervous system’s protective responses against insults, partly due to its antioxidant properties and its ability to inhibit cell death pathways [64]. Studies have demonstrated that overexpressed HSP70 and HSP27 work together in varying proportions to reduce demyelination, thereby improving functional recovery in a mouse model of MS [65]. Both immune-mediated and neurodegenerative mechanisms undoubtedly contribute to the development of this disease. In the relapse phase of multiple sclerosis, elevated levels of HSP27 and HSP70 have been observed in the serum and cerebral fluids of patients [31,66]. These findings imply that HSPs might be upregulated during MS attacks as a protective response to prevent protein misfolding and aggregation, ultimately safeguarding neuronal cells [28]. The exact role and significance of HSPs in MS pathology and disease progression are still not fully understood. It is possible that both HSP70 and HSP27 dysregulation in MS could be a consequence of the complex interplay between inflammation, immune response, and cellular stress in the disease.

Cerebrospinal fluid (CSF) is secreted from various central nervous system (CNS) structures, and any changes in its protein composition may signal altered brain protein expression, particularly in neurodegenerative and other CNS disorders. Cerebrospinal fluid (CSF) is a more precise medium for biomarker discovery in neurological diseases like MS or Alzheimer’s disease due to its proximity to the CNS, but its collection is invasive and associated with potential risks compared to plasma sampling. As a result, CSF samples from healthy individuals are rarely obtained, particularly for discovery-driven research. Conversely, blood plasma, easily collected through a non-invasive procedure, can serve as a promising medium for identifying diagnostic biomarkers by reflecting the pathological and physiological state of various tissues and blood cells. Additionally, mRNA levels in plasma and CSF may also differ. Plasma mRNA profiles can provide systemic information, while CSF mRNA profiles may better reflect gene expression changes specific to the affected CNS regions [67].

## 4. Materials and Methods

### 4.1. Materials

In this study, 45 patients with diagnosed relapsing-remitting multiple sclerosis (RRMS), comprising 27 females and 18 males (aged 56 ± 4.5 years), were recruited from the Neurological Rehabilitation Division at the III General Hospital in Lodz. All patients diagnosed with relapsing-remitting multiple sclerosis (RR-MS) were in the stable phase of the disease, experiencing remission without any recent attacks. None of the MS patients included in this study had received any form of immunomodulatory therapy within the three months leading up to the collection of blood samples. The diagnosis of MS in patients was based on the latest McDonald’s criteria (2017 version). Furthermore, 48 healthy individuals without MS, comprising 28 females and 20 males (aged 54 ± 5.5 years), were recruited for this study from the Vadimed Medical Center in Krakow, Poland (Table 1).

All the participants enrolled in this study, including both patients and control subjects, were Caucasians. The study design received approval from the Committee for Bioethics of the Medical University of Lodz in Poland and adhered to the principles outlined in the Declaration of Helsinki. Before the study commenced, informed written consent was obtained from all participants after a thorough explanation of the study. The clinical diagnosis of multiple sclerosis (MS) in patients was determined based on the McDonald’s criteria (2017 version). Several exclusion criteria were applied, including age below 18 years or above 70 years, severe general health condition of the patient, the presence of another neurological disease or autoimmune disease, the presence of cancer, inability to provide informed consent due to difficulties in logical verbal communication, the presence of severe psychiatric illness hindering informed consent, and active inflammatory acute disease. Prior to the examination, patients with multiple sclerosis did not exhibit any additional inflammatory diseases or cancer at the time of blood collection.

### 4.2. Methods

#### 4.2.1. Blood Sample Collection, RNA Preparation, and Reverse Transcription Quantitative Real-Time PCR RNA: Isolation and cDNA Synthesis

Nine milliliters of blood was drawn via venepuncture into S-Monovette^®^ K3 EDTA, cap red, between 8 and 9 am. Total RNA was isolated from PBMC using a GeneMATRIX Human Blood RNA Purification Kit, according to the manufacturer’s protocol, no later than 2 h after sample collection. Total RNA was extracted from 42 patients with MS and 48 subjects without MS. RNA was eluted in 50 μL RNase-free water and stored at −20 °C. RNA quality and quantification were measured spectrophotometrically using a Synergy/HT spectrophotometer and software, applying the 260/280 and 260/230 ratio algorithms. RNA with a 260/280 nm ratio in the range of 1.8–2.0 was considered high quality and was used for further analysis. cDNA was synthesized from 400 ng of total RNA with a High-Capacity cDNA Reverse Transcription Kit (Applied Biosystems™, Thermo Fisher Scientific, Waltham, MA, USA) following the manufacturer’s protocol. The cDNA was subjected to quantitative real-time PCR using the CFX Connect Real-Time System (BioRad, Hercules, CA, USA) with TaqMan PCR Master Mix and TaqMan Gene Expression Assays (Applied Biosystems, Waltham, MA, USA) for BDNF, NT4/5, SIRT1, HSP70, HSP27, and GAPDH mRNA (BDNF Hs02718934_s1, NT4/5 Hs01921834_s1, SIRT1 Hs01009005_m1, HSP70 Hs04187663_g1, HSP27 Hs03044127_g1, and GAPDH Hs03929097_g1), which were used according to the manufacturer’s instruction. The *GAPDH* gene was used as the internal sample control. All samples were analyzed in duplicate, and in the event of a discrepancy (Ct), the result was rejected, and the level of expression of a given gene in a given patient was analyzed again. Ct value was determined by the number of cycles needed to exceed the background signal. The abundance of mRNAs in the studied material was quantified using the 2^−ΔCt^ method.

#### 4.2.2. Plasma Sample Collection, ELISA Test

Blood was drawn via venepuncture, collecting 9 mL into an S-Monovette^®^ K3 EDTA tube with a red cap between 8 and 9 am. Within one hour of collection, the dedicated tubes were processed to separate cells from plasma. This was achieved via centrifugation in pyrogen/endotoxin-free tubes for 10 min at 1500× *g*, using a refrigerated centrifuge at 4 °C. After plasma collection, samples were aliquoted onto sterile 96-well plates (pyrogen/endotoxin-free) dedicated to each analyzed protein to avoid multiple freeze–thaw cycles. The plasma aliquoted onto the plates was immediately frozen at a temperature of −20 °C.

The fundamental principle of ELISA involves the specific binding between an antigen (the molecule being detected) and an antibody (a molecule that specifically binds to the antigen). The “enzyme-linked” part of the assay’s name refers to the enzyme that is attached to the secondary antibody. This enzyme, when provided with its substrate, facilitates a color change reaction. The intensity of this color change is proportional to the amount of antigen in the sample, and it can be measured using a spectrophotometer.

The choice of ELISA type and specific setup often depends on the nature of the sample, the target molecule, and the specific requirements of sensitivity and specificity. In this research, Sandwich ELISA Kits by Thermo Fisher Scientific Inc (Life Technologies Corporation, Carlsbad, CA USA). were used for all proteins, and all test procedures were conducted following the manufacturer’s protocol. Absorbance was measured using the Thermo Scientific™ Multiskan™ FC Microplate Photometer, and protein concentration was determined by calculating the average absorbance of each sample based on the standard curve dedicated to each measured protein.

#### 4.2.3. Statistical Analysis

All statistical analyses were conducted with Prism 8 (GraphPad Software, San Diego, CA, USA). Data are presented as means ± SEM (standard error of the mean) of the conducted experiments. The distribution of variables was evaluated using the Shapiro–Wilk test and statistical analysis of differences between the groups of data was carried out using the Mann–Whitney U test (for non-normal distribution). Values of *p* < 0.05 were regarded as statistically significant (*p* * ≤ 0.05, *p* **≤ 0.01, and *p* **** ≤ 0.0001).

## 5. Conclusions

In this comprehensive study of multiple sclerosis (MS), we observed significant alterations in gene expression and corresponding protein levels in MS patients compared to healthy controls. Moreover, our investigation revealed a remarkable synchronicity between gene expression and protein concentration, strengthening the notion that these neuroprotective genes play pivotal roles in MS-related processes. This research reveals the complexity of MS, stemming from interactions between genetics and the environment. It emphasizes the need to monitor genetic and protein-level changes for potential therapeutic interventions. Our study contributes to understanding MS and suggests promising paths for future research and treatment.

## Figures and Tables

**Figure 1 ijms-24-16176-f001:**
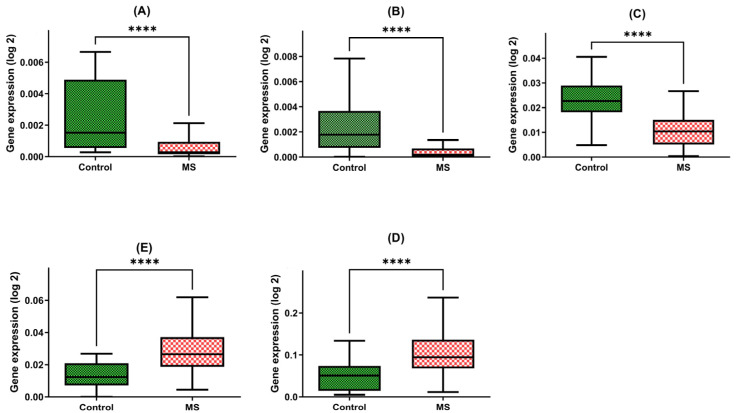
Expression levels of *BDNF* (**A**), NT-4 (**B**), *SIRT1* (**C**), *HSP70* (**D**), and *HSP27* (**E**) in PBMCs from MS patients and healthy controls. Statistical analysis of differences between the groups of data was carried out using the Mann–Whitney U test (**** indicates statistical significance at a *p* < 0.0001).

**Figure 2 ijms-24-16176-f002:**
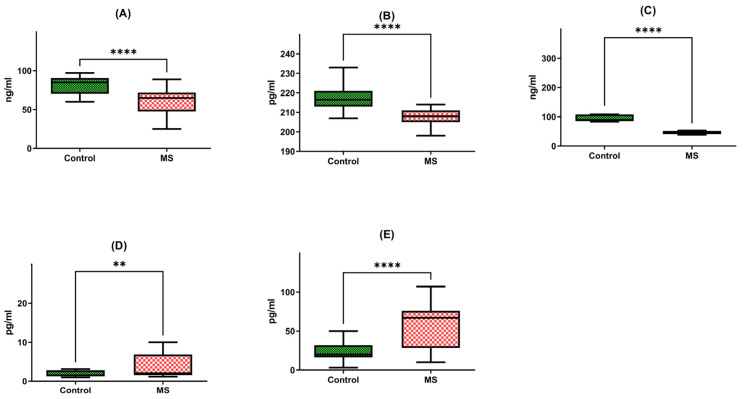
Protein concentrations of BDNF (**A**), NT-4 (**B**), SIRT1 (**C**), HSP70 (**D**), and HSP27 (**E**) in PBMCs from MS patients and healthy controls. Statistical analysis of differences between the groups of data was carried out using the Mann–Whitney U test (**** indicates statistical significance at a *p* < 0.0001, ** indicates statistical significance at a *p* < 0.01).

**Figure 3 ijms-24-16176-f003:**
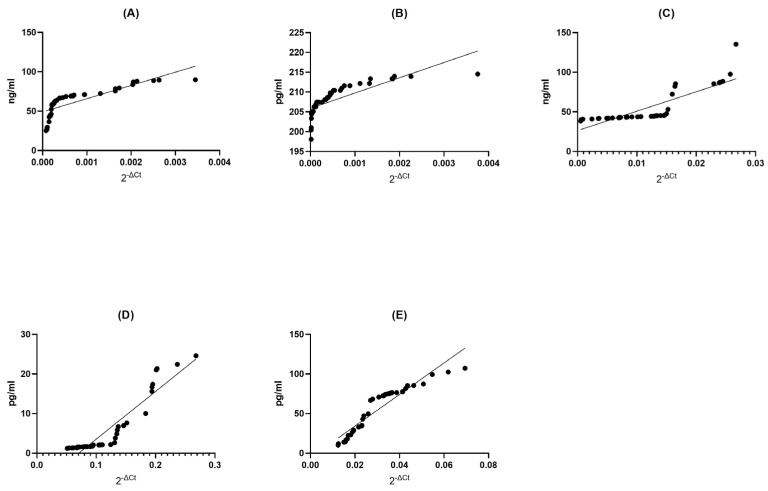
Assessment of correlations between gene expression and protein concentration of BDNF (**A**), NT-4 (**B**), SIRT1 (**C**), HSP70 (**D**), and HSP27 (**E**) in PBMCs from multiple sclerosis patients. Statistical significance was analyzed using Spearman’s rank correlation test (*p* < 0.0001).

**Figure 4 ijms-24-16176-f004:**
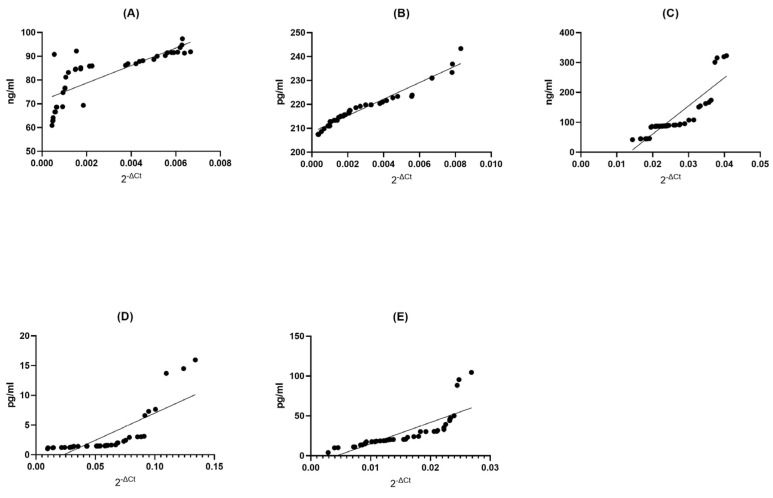
Assessment of correlations between gene expression and protein concentration of BDNF (**A**), NT-4 (**B**), SIRT1 (**C**), HSP70 (**D**), and HSP27 (**E**) in PBMCs from healthy volunteers. Statistical significance was analyzed using Spearman’s rank correlation test (*p* < 0.0001).

**Table 1 ijms-24-16176-t001:** Clinical characteristics of the 42 patients with relapsing-remitting multiple sclerosis (RRMS) and control group; 10–5.5—no or little impairment to walking; 6–6.5—requires one or two walking aids; >7—wheelchair mobility or confined to bed.

	MS Group	Control Group
Number of subjects	45	48
Females/males	27/18	28/20
Age (years, mean +/−SD)	56/4.5	54/5.5
Expanded disability status scale (EDSS) at the stable phase (range)	5.5 ± 1.0	-

## Data Availability

Data are contained within the article.

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
