# Peer review of "Exploring the mRNA and Plasma Protein Levels of BDNF, NT4, SIRT1, HSP27, and HSP70 in Multiple Sclerosis Patients and Healthy Controls"

_ijms, 2023, doi:10.3390/ijms242216176_

Round 1

Reviewer 1 Report

Comments and Suggestions for Authors

This is a valuable study based on gene and protein expression analysis.

I have some comments to help to improve the manuscript:

1. Did the authors perform an analysis of the MS activity? If the data are available, I recommend the authors add MS activity and biomarker analysis, too;

2. Under the methods section, the explanation of ELISA testing is too elaborate and can be shortened by removing the basic explanation of the ELISA test types and principles. This is unnecessary in this part and most of the readers should be aware of the details of ELISA testing;

3. The English needs some corrections, especially checking the wording. For instance, the word "showcase" was used in the wrong context.

Comments on the Quality of English Language

The English needs some corrections, especially checking the wording. For instance, the word "showcase" was used in the wrong context.

Author Response

Dear reviewer.
Thank you for your valuable feedback and guidance. Regrettably, I currently don't possess the requested data. However, we are planning to continue our research in this area in the future. Incorporating MS activity data will undoubtedly enhance our research and contribute to a more comprehensive exploration in our upcoming work.
Best regards 
Igor Sokolowski 

Reviewer 2 Report

Comments and Suggestions for Authors

The manuscript of Sokolowski et al. analyzes the gene expression and protein levels of the selected genes in a local MS cohort, composed of 42 MS patients, compared to healthy controls. The selected genes investigated on the mRNA and protein levels were: BDNF, NT4/5, SIRT1, HSP70, and HSP27. The authors found differences in mRNA expression and in protein levels encoded by these genes in the MS cohort in comparison to controls. Also, the mRNA expression correlated well with protein levels.

The introduction is well written and provides the necessary background on the factors that may be a risk factor for developing MS. Figures are illustrative and of good quality. Methods are described in detail. There are a few issues that could be added to the discussion (please see below). The cohort is quite small and only from one center.

My main questions relate to the facts of how the genes have been selected and if any other genes, also not showing significant differences, in the cohort have been described.

Comments:

1. My main question is how the genes have been selected, given that the information on these gene expressions and protein levels in the literature is inconclusive. Have any other genes, not mentioned in the text, been investigated on the mRNA or protein level and were not significant? If yes, please mention or add to the supplement.

2. The title is misleading as the authors do not investigate neuroprotective effects, but rather the mRNA and protein levels of the selected genes in plasma in the MS cohort and compare with healthy controls

3. Line 51 "Changes in the IL7R gene" - is overexpression meant or some specific rs?

 4. Line112-113 "protein levels as potential biomarkers for assessing inflammatory processes and overall disease progression"- has this been assessed and how?

5. Line 115 I am not sure if they can already be named “biomarkers”

 6. Please write the names of the genes in italics throughout the text

7. Discussion- Please comment on how the protein and mRNA levels in MS in blood relate to the one in CSF. Also, comment about other diseases (CSF vs. plasma) e.g. AD

 8. It would be good to present the disused results of the mRNA and protein levels in other studies in the table (e.g. in the supplement) and present also the size of the cohort and methodology applied. Did all of these studies perform measurements in plasma?

 9. Are there any data on how long the patients in the local cohort had MS?

 10. Table 1 and cohort characteristics may be included in the results.

 11. Have any other methods, such as proteomics, been considered?

 12. The second part of the conclusions is wordy

Author Response

Dear reviewer,
thank you for your comments and advices. I will be happy to answer all your questions one by one.
1. Our gene selection process was performed by a thorough analysis of existing scientific literature. Nevertheless, like yourselves, we have also noticed that the reports regarding the correlation between mRNA levels and gene expression are highly inconclusive, and the limited number of available reports currently prevents us from conducting a meta-analysis. At present, drawing definitive conclusions about the extent of mRNA level influence on protein concentration isn't straightforward. Essentially, our goal was to identify a group of genes whose activity might play a significant role in the overarching mechanism of neuroprotection. In the near future, we are preparing to investigate the correlations between mRNA levels and protein concentrations for key genes, including HSP90, HSP10, HSP60, NT3, and NGF.
2. Our objective in scrutinizing the activity of these selected genes is to assess the extent of Multiple Sclerosis's influence on the neuroprotection mechanism. We've selected protein concentration as the measure of effectiveness for our assessment. Consequently, we've titled our article "Neuroprotective Effects of BDNF, NT4/5, SIRT1, HSP70, and HSP27 Genes in Multiple Sclerosis Patients." This title reflects our focus on how these specific genes may impact neuroprotection in individuals with Multiple Sclerosis, with a particular emphasis on protein concentrations.
3. I have expanded this part. "Changes in the IL7R gene are also associated with an increased risk of developing MS. The specific changes caused by the genetic variant rs6897932 in the IL7R gene are related to alterations in the regulation and function of the interleukin-7 receptor (IL-7R). The exact mechanisms by which rs6897932 and similar variants contribute to an increased risk of multiple sclerosis (MS) are still being studied, but it is believed that these genetic changes may lead to dysregulation of the immune system"
4. In multiple sclerosis (MS), clinical changes often involve inflammation and demyelination, key characteristics of the disease. Research suggests that alterations in neuroprotective gene expression and protein levels can result directly from MS-related inflammation. While we haven't investigated the correlation between inflammation markers, mRNA, and protein levels at this time, your suggestion is valuable and could be a promising direction for future research.
5. I agree. Removing "biomarker" from the text was a good decision.
6. Done
7. 

MS: Plasma vs. CSF:

  Proteins: In MS, protein biomarkers in plasma and CSF may show differences. While certain inflammatory or neurodegenerative markers can be elevated in both, the levels in CSF are often more indicative of CNS-specific changes, as CSF is in closer proximity to the brain and spinal cord. Proteins like oligoclonal bands and neurofilament light chain are commonly used CSF biomarkers.

  mRNA: mRNA levels in plasma and CSF may also differ. Plasma mRNA profiles can provide systemic information, while CSF mRNA profiles may better reflect gene expression changes specific to the affected CNS regions. Studying these differences can help understand the disease's molecular mechanisms.

Alzheimer's Disease (AD): CSF vs. Plasma:

  Proteins: In AD, protein biomarkers such as amyloid-beta and tau protein are often assessed in both CSF and plasma. However, their levels in CSF are typically more indicative of disease progression because CSF is closer to the Plasma levels may be influenced by systemic factors.

  mRNA: mRNA profiles in CSF and plasma have also been examined in AD. Differences in mRNA expression may offer clues about disease mechanisms.

Comparing protein and mRNA levels in CSF and plasma is valuable for understanding the specific and systemic aspects of disease pathology. While CSF provides a closer look at CNS-related changes, plasma samples are more accessible and can provide systemic information. The choice of sample type depends on the disease and the specific research or diagnostic goals. In the case of AD, CSF is often used for more direct insights into CNS pathology, while plasma may be used for broader population-based studies. The right choice of sample depends on the research objectives and the stage of the disease under investigation.

8. Currently, I do not have the additional results regarding mRNA and protein levels for the genes discussed in the article, nor aditional ones which were not mentioned in the paper. In close future we want to perform new tests for HSP90, HSP10, HSP60, NT3, and NGF.

9. Currently, I don't have access to this data.

10. In a scientific original paper, details regarding the cohort's characteristics are usually found in the "M&M" section. However, I personally find it more suitable to incorporate information about the cohort within the "M&M" section. Nevertheless, if it is preferred to place this information in the "Results" section, I will do it.

11. In the future, who knows? Currently, proteomics is not within the scope of our department's activities.

12. Done.

Please see an attachement.

Thank you once again for your comments and advice.

Kindest regards

Igor SokoÅ‚owski 

Round 2

Reviewer 2 Report

Comments and Suggestions for Authors

Dear authors,

Thank you for the corrections. My main objection is the size of the cohort and the limited number of proteins investigated. 

I believe that it's a well-written manuscript, the results are presented clearly and the research part is performed correctly. Still, I would change the title, because issues, such as the mechanism of neuroprotection or correlation with clinical measures and protein levels have not been investigated. I would add a few sentences regarding plasma and CSF differences for protein and mRNA concentration either to the discussion or to the limitations part. It may be that I missed this in the text: for the reviewer, it would be easier if you could work in the track changes mode.

Author Response

Dear Reviewer,

I would like to express my gratitude for your valuable comments and advice. As is customary, I will address your comments one by one:1. Addressing your main objection, I'd like to clarify that the choice to focus on proteins rather than both mRNA and proteins was influenced by the limited availability of studies that describe both mRNA and protein levels in the context of multiple sclerosis (MS). Such correlations are indeed scarce in the literature, which made it challenging to base our research on such comprehensive data.

To put it into perspective, when analyzing existing studies that specifically investigated neurotrophin levels, I observed that many of them focused on just a few neurotrophins (usually 1 or 2) and employed relatively small cohorts. For example, Wens et al. used a cohort of 22 MS patients, with an associated journal impact factor (IF) of 5.1. Petereit et al. worked with 28 MS patients and published their results in a journal with an IF of 5.855, while Sarchielli et al. used a cohort of 60 MS patients for their publication. Describing SIRT1, Emamgholipour et al. had a cohort of 14 patients and an associated journal IF of 6.78. Pennisi et al. (2011) worked with 26 MS patients, describing five proteins, including HSP70 and SIRT1, and their work was published in a journal with an IF of 6.1. Gorter et al. analyzed six chaperones with a cohort of 20 patients and an associated journal IF of 5.73.

Given the information from these previously published studies, I'd like to emphasize that while a larger cohort is always beneficial, the cohort in our research included 42 MS patients and 48 healthy control subjects. Importantly, the results obtained from our cohort have demonstrated high statistical significance. It's worth noting that the scarcity of data describing both mRNA and protein concentrations in MS influenced our approach.

In my humble opinion, our research meets the requirements of the International Journal of Molecular Sciences, which you represent. I do believe that our article provides valuable insights into the field of MS, even when compared to other studies published in the journal.

Citations
Emamgholipour, S., Hossein-Nezhad, A., Sahraian, M. A., Askarisadr, F., & Ansari, M. (2016). Evidence for possible role of melatonin in reducing oxidative stress in multiple sclerosis through its effect on SIRT1 and antioxidant enzymes. Life Sciences, 145, 34–41. https://doi.org/10.1016/j.lfs.2015.12.014

Gorter, R. P., Nutma, E., Jahrei, M. C., de Jonge, J. C., Quinlan, R. A., van der Valk, P., van Noort, J. M., Baron, W., & Amor, S. (2018). Heat shock proteins are differentially expressed in brain and spinal cord: implications for multiple sclerosis. Clinical and Experimental Immunology, 194(2), 137–152. https://doi.org/10.1111/cei.13186

Pennisi, G., Cornelius, C., Cavallaro, M. M., Salinaro, A. T., Cambria, M. T., Pennisi, M., Bella, R., Milone, P., Ventimiglia, B., Migliore, M. R., Di Renzo, L., De Lorenzo, A., & Calabrese, V. (2011). Redox regulation of cellular stress response in multiple sclerosis. Biochemical Pharmacology, 82(10), 1490–1499. https://doi.org/10.1016/j.bcp.2011.07.092

Petereit, H. F., Lindemann, H., & Schoppe, S. (2003). Effect of immunomodulatory drugs on in vitro production of brain-derived neurotrophic factor. Multiple Sclerosis, 9(1), 16–20. https://doi.org/10.1191/1352458503ms869oa

Sarchielli, P., Zaffaroni, M., Floridi, A., Greco, L., Candeliere, A., Mattioni, A., Tenaglia, S., Di Filippo, M., & Calabresi, P. (2007). Production of brain-derived neurotrophic factor by mononuclear cells of patients with multiple sclerosis treated with glatiramer acetate, interferon-β 1a, and high doses of immunoglobulins. Multiple Sclerosis, 13(3), 313–331. https://doi.org/10.1177/1352458506070146

Wens, I., Keytsman, C., Deckx, N., Cools, N., Dalgas, U., & Eijnde, B. O. (2016). Brain derived neurotrophic factor in multiple sclerosis: Effect of 24 weeks endurance and resistance training. European Journal of Neurology, 23(6), 1028–1035. https://doi.org/10.1111/ene.12976

2. You made a good suggestion to change the article's title. The new title, "Exploring the mRNA and Plasma Protein Levels of BDNF, NT4, SIRT1, HSP27, and HSP70 in Multiple Sclerosis Patients and Healthy Controls," hope the new title provides a clearer and more comprehensive description of the scope of our research.

3. I have added a few sentences about plasma and CSF. Please find in the attachment. 

Thank you once more for your feedback and guidance. I am optimistic that, after this round of revisions, my article will align with all your requirements and expectations.

Kindest regards 

Igor SokoÅ‚owski 
